# DP-MIA: Dual-Phase Membership Inference Attack across VLMs Training Lifecycle

## Abstract

Recent advancements in Vision-Language Models (VLMs) have amplified privacy concerns of training data source attribution, due to their multi-stage training lifecycle and growing deployment via black-box APIs. The SOTA source attribution approach - Membership Inference Attacks (MIAs) primarily focus on coarse-grained binary classification, oversimplifying the complex exposure risks in multi-stage training. Crucially, MIAs fail to differentiate whether data was exposed during pretraining or finetuning, hindering precise accountability tracing in real-world VLM development. To bridge this gap, we introduce DP-MIA (Dual-Phase Membership Inference Attack), a novel framework that uniquely distinguishes across three exposure states: pretrain-member, finetune-member, or non-member. This multi-class formulation captures fine-grained privacy risks across distinct training stages, enabling significantly more precise source attribution. Designing DP-MIA presents two key challenges: limited model access (black-box setting) and subtle memorization signals. We tackle these challenges through three novel strategies: 1) Cosine Similarity Attack (CSA): exploits semantic alignment shifts between phases; 2) RIGEL-based Multi-class Classifier: leverages a new composite metric (RIGEL) integrating generation response time, inference confidence and generation length for enhanced signal detection; 3) Dual-Binary Attack (DBA): decomposes the inference hierarchically into two binary sub-problems. Extensive experiments on LLaVA and Qwen2-VL demonstrate DP-MIA's effectiveness (88.2% accuracy) significantly outperforming baselines such as CSA and MCA. Our findings expose critical vulnerabilities in VLM training pipelines and provide actionable insights for privacy auditing in black-box scenarios. DP-MIA's code is available at `https://github.com/frozen-jak/Dual-Phase-Mia`.

## 1 Introduction

Recent advancements in vision-language models (VLMs) such as CLIP Radford et al. (2021), BLIP Li et al. (2022), Qwen2-VL Inc. (2024), LLaVA Liu et al. (2023b) have dramatically enhanced machines' ability to understand and generate multimodal content. However, this progress hinges on large-scale and often sensitive training data, raising critical privacy concerns. Similar to language models Carlini et al. (2021); Hayes et al. (2019), VLMs are vulnerable to membership inference attacks (MIAs) Shokri et al. (2017); Yeom et al. (2018), where adversaries determine if specific data was used in training. Successful MIAs risk exposing proprietary information or personal data.

Vision-language models are generally trained in two stages: a pretraining phase and a finetuning phase. In the pretraining phase, the model learns to align image-text pairs from large-scale datasets (e.g., LAION Schuhmann et al. (2021)), capturing rich semantic correspondences between visual and linguistic signals. The finetuning phase further adapts the model to follow user instructions or perform downstream tasks by training on smaller, curated datasets Liu et al. (2023a); Zhou et al. (2022). These stages often differ not only in objectives and data sources but also in organizational ownership and privacy responsibilities. Both stages are susceptible to memorization that can expose training data Carlini et al. (2022); Song et al. (2023). This creates a key accountability challenge—when a model reveals signs of data leakage, it is unclear whether the memorization originated from pretraining or finetuning stage. Identifying the source stage of such leakage is essential for auditing model behavior, assigning responsibility, and ensuring compliance with data governance regulations Strubell et al. (2019); Kravitz et al. (2023).

In this work, we investigate a novel membership inference attack setting targeting multi-stage Vision-Language Models (VLMs). Our objective extends beyond traditional binary classification: we aim to determine not only if a sample was used in training, but precisely when - distinguishing whether exposure occurred during pretraining, finetuning, or not at all. This fine-grained approach reflects real-world deployment where multiple parties contribute data at different stages, making precise accountability attribution essential Tramèr et al. (2022); Sun et al. (2019). Unfortunately, existing MIA methods—while effective against traditional ML models and LLMs Shokri et al. (2017); Carlini et al. (2021); Song et al. (2021) - perform poorly on VLMs and LLMs for two key reasons: 1) **Massive Scale**: Training involves trillions of tokens/billions of image-text pairs Brown et al. (2020); Radford et al. (2021), diluting memorization signals; 2) **Black-Box Constraints**: API-only access severely limits attack capabilities Wu et al. (2022); Zhu et al. (2023).

To address these challenges, we propose a novel membership inference framework that exploits subtle behavioral signals in VLMs when processing different data types. Moving beyond conventional output similarity checks, our approach targets consistent patterns induced by overfitting during training. This foundation yields three key designs: 1) **Composite Metric** - We introduce RIGEL (Response time, Inference confidence, Generation Length) – a novel metric integrating three observable black-box signals: response generation time, model confidence score, and output length, inspired by recent blackbox VLMs findings Melis et al. (2019); Prabhudesai et al. (2025); Carlini et al. (2022); Jiang et al. (2023); 2) **Group-Based Inference**- We amplify overfitting signals by analyzing batches of related samples, inspired by privacy auditing and medical analysis techniques Li et al. (2022); Sablayrolles et al. (2019), which enhances detection robustness. 3) **Phase-Specific Attribution**- We explicitly differentiate between pretraining and finetuning membership – a critical advancement for accountability-focused privacy threats Liu et al. (2023a); Zhou et al. (2022).

These components form our attack suite: a Cosine Similarity Attack (CSA) exploiting semantic alignment shifts, a RIGEL-based classifier, and a hierarchical Dual-Binary Attack (DBA) implementing phase distinction. Our main contributions are summarized as follows: 1) **Pioneering Multi-Class MIA for VLMs**: We introduce the first membership inference framework that distinguishes pre-training members, instruction-tuning members, and non-members in multi-stage VLMs; 2) **Composite Black-box Metric**: We propose RIGEL, a novel composite metric leveraging black-box observable signals (response time, confidence, output length) to detect subtle memorization patterns; 3) **Hierarchical Attack Design**: We design a Dual-Binary Attack (DBA) with grouped inference, enhancing robustness against black-box constraints via hierarchical decision-making and signal amplification; 4) **Rigorous Validation**: Through extensive experiments on LLaVA and Qwen2-VL—including ablation studies—we demonstrate the efficacy of our multi-stage approach, group inference, and RIGEL features (DBA achieves 88.2% accuracy).

## 2 RELATED WORK

### 2.1 MEMBERSHIP INFERENCE ATTACKS (MIA)

Membership Inference Attacks (MIAs) aim to determine if a specific data sample was used to train a machine learning model. Early MIA research primarily targeted traditional supervised models—including deep neural networks Shokri et al. (2017) and generative models Hayes et al. (2019); Hu et al. (2021)—demonstrating that adversaries can exploit model overfitting to infer membership with success rates significantly exceeding random chance. However, MIAs face substantially greater challenges against large-scale models like LLMs and VLMs. The immense scale and diversity of their training data—trillions of tokens for LLMs Brown et al. (2020) and billions of image-text pairs for VLMs Radford et al. (2021); Alayrac et al. (2022)—result in minimal per-sample overfitting. Consequently, MIA techniques effective against traditional models frequently fail for LLMs/VLMs, as individual samples exert negligible influence on global model behavior Carlini et al. (2021).

### 2.2 MIA IN VISION-LANGUAGE MODELS

Recent studies have extended MIA research to Vision-Language Models (VLMs). Related work Yu et al. (2022) focused on the finetuning stage of VLMs, demonstrating that finetuned models are susceptible to membership inference. However, their approach is limited as it only considers the finetuning phase, and does not address the challenge of attacks across both pretraining and fine-

tuning stages. This limitation is crucial, as many VLMs are trained in multiple stages, with data contributions from different parties at each stage. Thus, attacks on multi-stage VLMs require a more sophisticated approach to distinguish between samples used in the pretraining and finetuning phases. Additionally, the black-box nature of modern VLMs poses another significant challenge. Black-box models are typically accessed via APIs that do not expose hyperparameters or other critical internal details. This limited access complicates the application of traditional MIA methods, which often rely on detailed model internals to identify overfitting behavior. Recent work on black-box attacks has explored new strategies Yeom et al. (2018); Salem et al. (2018), yet the challenge of applying these methods to multi-stage training remains an open problem.

## 2.3 PRIVACY RISKS IN PRETRAINING AND FINETUNING

The two-stage training pipeline of VLMs introduces distinct privacy risks at each phase. Pretraining typically involves large, diverse public datasets, whereas finetuning often uses smaller, task-specific datasets that may contain more sensitive information. Consequently, identifying whether privacy leakage originates from pretraining or finetuning is crucial for assigning responsibility and guiding data governance policies. Existing MIA research has largely focused on either pretrained or finetuned models in isolation, without addressing the increasingly common scenario where VLMs undergo multi-stage training involving both pretraining and finetuning phases. Each stage may involve distinct datasets, data owners, and privacy constraints, making it imperative to accurately identify the origin of data leakage for effective accountability and compliance.

## 3 PRELIMINARIES

**Vision-Language Model Architecture**: VLMs generally follow a two-stream architecture, with one stream processing visual data (images) and the other processing textual data (texts). Both streams are projected into a shared high-dimensional embedding space where cross-modal reasoning can be performed. Let $\mathbf{x}_{\text{img}} \in \mathbb{R}^{d_{\text{img}}}$ represent the visual input (an image), and $\mathbf{x}_{\text{text}} \in \mathbb{R}^{d_{\text{text}}}$ represent the textual input (a sentence or phrase). The objective of a VLM is to map these two different modalities into a shared representation space $\mathcal{H}$, such that each modality is embedded in this space and can be reasoned about jointly.

**Visual and Textual Embeddings**: The key challenge in VLMs is the alignment of visual and textual information in the shared embedding space. To achieve this, the model employs separate encoders for images and text, which are typically transformer-based architectures.

Visual Encoder processes the image $\mathbf{x}_{\text{img}}$ and produces an embedding $\mathbf{z}_{\text{img}} \in \mathbb{R}^{d_{\text{img}}}$:

$$\mathbf{z}_{\text{img}} = f_{\text{img}}(\mathbf{x}_{\text{img}})$$

Textual Encoder processes the text $\mathbf{x}_{\text{text}}$ and produces an embedding $\mathbf{z}_{\text{text}} \in \mathbb{R}^{d_{\text{text}}}$:

$$\mathbf{z}_{\text{text}} = f_{\text{text}}(\mathbf{x}_{\text{text}})$$

**Dual Stage Training**: VLMs are typically trained in two stages: pretraining and finetuning. The pretraining objective generally involves minimizing a contrastive loss function as described earlier, while finetuning typically involves supervised learning for specific downstream tasks, such as cross-entropy loss for classification or sequence-to-sequence loss for caption generation.

$$\mathcal{L}_{\text{fine-tune}} = -\frac{1}{N} \sum_{i=1}^{N} \log P(\hat{y}_i | \mathbf{x}_{\text{img},i}, \mathbf{x}_{\text{text},i}) \tag{1}$$

where $P(\hat{y}_i | \mathbf{x}_{\text{img},i}, \mathbf{x}_{\text{text},i})$ is the probability of correct output given the input image and text pair.

## 3.1 THREAT MODEL

We now outline the adversary's goal, assumptions, and available capabilities.

**Adversarial Objective**: Given a target Vision-Language Model (VLM), the adversary aims to classify a data instance into one of three categories: 1) Pretraining member: Used in pretraining phase;

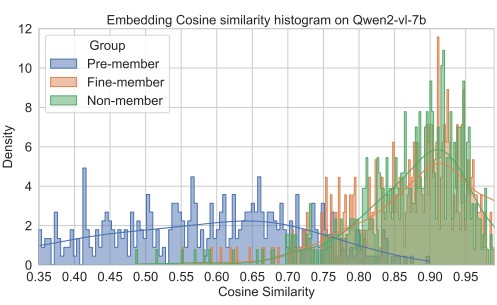 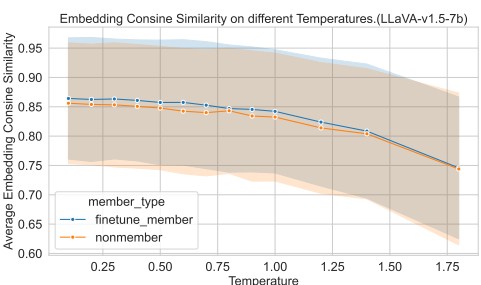

Figure 1: Cosine similarity histogram.  Figure 2: Different Temperature on LLaVA-7b.

2) Finetuning member: Used in instruction-tuning phase; 3) Non-member: Absent from both training stages. Classification relies exclusively on observable model outputs.

**Model Access**: We assume a strict black-box setting reflecting production deployments. The adversary: 1) cannot access internal arch/parameters or gradients; 2) can query the model with (image, text prompt, ground truth) triplets; 3) can only observe the generated responses.

**Adversarial Data Knowledge**: The adversary possesses a limited reference dataset with verified membership labels: 1) Pretraining members sampled from public pretraining datasets (e.g., LAION, CC, SBU Schuhmann et al. (2021); Sharma et al. (2018); Ordonez et al. (2011)); 2) Finetuning members curated from domain-relevant sources, filtered to minimize overlap with pretraining/non-members; 3) Non-members collected post-model-release to ensure temporal exclusion.

**Computational Constraints**: Attacks requiring VLM-scale shadow model training are computationally prohibitive due to GPU/memory demands. The adversary is therefore restricted to: a) Query-efficient strategies; b) Lightweight inference mechanisms.

## 4 METHODOLOGY

### 4.1 STRATEGY 1: COSINE SIMILARITY ATTACK (CSA)

The first key strategy is the *Cosine Similarity Attack (CSA)*, which exploits the overfitting behavior observed in Vision-Language Models (VLMs) Salem et al. (2019); Song et al. (2021). Our insight is that VLMs tend to produce more accurate and concise responses when queried with samples encountered during the finetuning stage. This results in a higher cosine similarity between the model's response and the ground-truth answer for these member samples. In contrast, the similarity score is typically lower for non-member samples. Interestingly, samples from the pretraining dataset (pretraining members) also exhibit relatively lower similarity scores. This is likely attributable to the alignment objective of pretraining, where image-text pairs are semantically aligned but are often less detailed than finetuned examples Li et al. (2023b); Alayrac et al. (2022).

Given a query sample $x = \{x_{\text{img}}, x_{\text{qst}}, x_{\text{ans}}\}$, where $x_{\text{img}}$ and $x_{\text{qst}}$ represent the image and prompt input, and $x_{\text{ans}}$ is the ground-truth answer, we denote the model's response as $y$. Let $f(\cdot)$ be a feature extractor (e.g., a sentence embedding model such as `all-mpnet-base-v2`) and $\text{CS}(\cdot, \cdot)$ the cosine similarity function. We define the membership inference function $M(x)$ as:

$$M(x) = \begin{cases} \text{Pretrain-member, if } \text{CS}(f(x_{\text{ans}}), f(y)) < \tau_1 \\ \text{Finetune-member, if } \text{CS}(f(x_{\text{ans}}), f(y)) > \tau_2 \\ \text{Non-member, if } \tau_1 \leq \text{CS}(f(x_{\text{ans}}), f(y)) \leq \tau_2 \end{cases}$$

where $\tau_1$ and $\tau_2$ are two similarity thresholds ($0 < \tau_1 < \tau_2$) selected based on a reference set. Figure 1 presents an example of such distribution-based separation.

### 4.2 STRATEGY 2: MULTI-CLASS MIA

Effective MIAs largely exploit a model's tendency to overfit its training data Yeom et al. (2018). As illustrated in Equation 1, Vision-language models (VLMs) exemplify this behavior, as their train-

---

**Algorithm 1** Multi-class Membership Inference Attack

---

**Input**: Query sample $x = \{x_{\text{img}}, x_{\text{qst}}, x_{\text{ans}}\}$; target model $f_\theta$; inference set $\mathcal{X}_{\text{inference}} = \{x_1, ..., x_n\}$
**Output**: Label {0:Pretrain-member, 1:Finetune-member, 2:Non-member}
1:  Extract features for each sample in $\mathcal{X}_{\text{inference}}$:
2:  **for** each $x_i = \{x_{\text{img},i}, x_{\text{qst},i}, x_{\text{ans},i}\} \in \mathcal{X}_{\text{inference}}$ **do**
3:      Query target model: $r_i \leftarrow f_\theta(x_i)$
4:      Extract answer embedding: $f_{\text{ans},i} \leftarrow \text{Embed}(x_{\text{ans},i})$
5:      Compute cosine similarity: $\text{CS}_i \leftarrow \text{sim}(r_i, f_{\text{ans},i})$
6:      $T_i \leftarrow$ *Per-embedding Generation Time*
7:      $C_i \leftarrow$ *Confidence Score*
8:      $L_i \leftarrow$ *Generated Embedding Length*
9:      Compute Rigel score: $\text{Rigel}_i \leftarrow \omega_1 T_i + \omega_2 C_i + \omega_3 L_i$
10:     Form feature vector: $v_i = [\text{CS}_i, \text{Rigel}_i]$
11: **end for**
12: Train multi-class classifier $\mathcal{C}$ using labeled feature vectors $\{v_i\}$
13: Compute feature vector $v$ for query sample $x$
14: Predict label $M(x) = \mathcal{C}(v)$
15: **return** $M(x) \in \{0, 1, 2\}$

---

ing objective explicitly aligns image and text inputs from training samples within the embedding space Radford et al. (2021); Li et al. (2023a). This alignment suggests a fundamental attack strategy: computing the cosine similarity between the model's output $y$ and the ground-truth answer $x_{\text{ans}}$. Samples seen during training are expected to yield higher similarity scores. However, this basic cosine similarity approach faces significant challenges with large-scale VLMs. Their remarkable generalization capabilities, combined with the relatively limited number of fine-tuning epochs, mean that even non-member samples frequently generate responses with cosine similarity scores comparable to those of fine-tuning members. This overlap makes reliable distinction difficult. Further complicating matters, recent research has investigated modulating VLM output behavior via the temperature parameter Yu et al. (2022). While empirical results confirm that increasing temperature does influence cosine similarity (Figure 2), the sensitivity to temperature changes remains subtle and shows minimal difference between finetuning-members and non-members. Consequently, distinguishing membership types based solely on variations in temperature and the resulting cosine similarity proves extremely challenging.

To overcome this challenge, we explore richer discriminative features beyond cosine similarity and propose a novel composite metric. This metric is specifically designed to capture the nuanced variations in model behavior across different membership types. We name this metric **RIGEL** - **R**unning Time, Conf**I**dence, and **G**enerated **E**mbedding **L**ength:

*Per-embedding Generation Time*: Execution time can reflect the model's uncertainty: longer generation time may imply the model is "thinking harder" when faced with unfamiliar inputs. To normalize this across varying response lengths, we define the average time per embedding token as:

$$T = \frac{1}{N} \sum_{i=1}^{N} t_i$$

where $t_i$ is the generation time for the $i$-th token $f(y)_i$ in the model's response $y$.

*Confidence Score*: Traditional MIA methods often rely on internal logits, which are inaccessible in black-box settings. However, LLM self-confidence can in-fact be queried explicitly. Inspired by real-world deployments, we append prompts like: *"Additionally, please rate your confidence to your answer on a scale from 0 to 100"*, to extract confidence scores even from proprietary models:

$$C = \psi(x)$$

where $x$ denotes the input sample (image and question), and $\psi(x)$ represents the model-reported confidence score in response to the appended natural-language query.

*Generated Embedding Length*: Prior research Carlini et al. (2023) has observed that LLMs tend to generate longer and more verbose answers with non-member inputs, due to the lack of strong

---

**Algorithm 2** Dual-Binary Membership Inference Attack

---

**Input**: Query sample $x = \{x_{\text{img}}, x_{\text{qst}}, x_{\text{ans}}\}$; target model $f_\theta$; threshold $\tau$
**Output**: Label {Pretrain-member, Finetune-member, Non-member}
 1: Query the target model: $y \leftarrow f_\theta(x)$
 2: Compute cosine similarity: CS $\leftarrow \text{sim}(f(x_{\text{ans}}), f(y))$
 3: **if** CS $< \tau$ **then**
 4:     **return** Pretrain-member
 5: **else**
 6:     Compute Rigel features:
 7:         $T \leftarrow$ *Per-embedding Generation Time*
 8:         $C \leftarrow$ *Confidence Score*
 9:         $L \leftarrow$ *Generated Embedding Length*
10:     Rigel $\leftarrow \omega_1 T + \omega_2 C + \omega_3 L$
11:     Predict membership using binary classifier $\mathcal{C}_{\text{Rigel}}$
12:     **return** $M(x) \in \{\text{Finetune-member}, \text{Non-member}\}$
13: **end if**

---

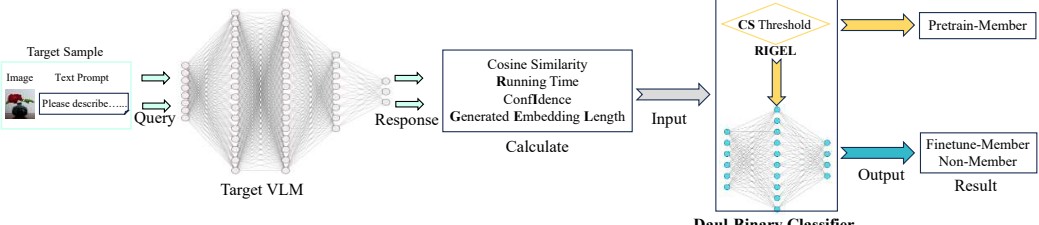

Figure 3: Overview of Dual-Binary Membership Inference Attack.

supervision signals. This is interpreted as the model "compensating" for its uncertainty. The length of the output embedding is thus another indicator:

$$L = \sum_{i=1}^{N} \|f(y)_i\|$$

Combining these components, the RIGEL score is:

$$\text{RIGEL}(x) = \omega_1 T + \omega_2 C + \omega_3 L$$

where $\omega_1, \omega_2, \omega_3$ are task-specific weights that can be optimized or manually tuned[1]. Finally, we train a classifier using both the RIGEL score and the cosine similarity to predict the membership type of an input sample. The full multi-class attack pipeline is summarized in Algorithm 1.

### 4.3 STRATEGY 3: DUAL-BINARY MIA

To further enhance membership inference precision in multi-stage Vision-Language Model (VLM) training pipelines, we propose a two-phase hierarchical strategy: the *Dual-Binary Membership Inference Attack (DB-MIA)*, as illustrated in Algorithm 2 and Figure 3. Unlike the multi-class classifier used previously, DB-MIA decomposes membership inference into two sequential binary classification sub-tasks. This hierarchical approach enables finer decision boundaries tailored to the distinct characteristics of each sample subgroup. The strategy's core intuition stems from differential sample behavior: 1) Pretrain-members typically generate shorter, less detailed responses with loose ground-truth alignment, making them readily separable from other samples using cosine similarity alone; 2) Finetune-members vs. non-members pose greater challenges, as both yield high similarity scores. Distinguishing these requires our more nuanced RIGEL metric. In Figure 1, cosine similarity shows a clear separation margin between pretraining members and others. Therefore, we design a two-stage decision process leveraging this divergence:

---

[1]Unless specified otherwise, we set $\omega_1 = 0.2$, $\omega_2 = 0.2$, and $\omega_3 = 0.6$ by default. This configuration is derived from the empirical results in the evaluation section, placing greater emphasis on the generated embedding length ($L$) compared to runtime ($T$) and confidence score ($C$).

Table 1: Performance comparison of three attack methods across four VLMs.

| Model | LLaVA-V7B | | | LLaVA-L7B | | | Qwen2-VL-7B | | | Qwen2-VL-2B | | |
|---|---|---|---|---|---|---|---|---|---|---|---|---|
| | Precision | Recall | Accuracy | Precision | Recall | Accuracy | Precision | Recall | Accuracy | Precision | Recall | Accuracy |
| Rényi(=5,max=1%) | 0.417 | 0.406 | 0.406 | 0.421 | 0.413 | 0.402 | 0.393 | 0.394 | 0.366 | 0.396 | 0.392 | 0.365 |
| CSA | 0.622 | 0.512 | 0.562 | 0.616 | 0.507 | 0.556 | 0.617 | 0.474 | 0.536 | 0.608 | 0.473 | 0.532 |
| MCA | 0.672 | 0.680 | 0.680 | 0.677 | 0.680 | 0.680 | 0.663 | 0.670 | 0.670 | 0.660 | 0.667 | 0.667 |
| DBA | 0.713 | 0.721 | 0.711 | 0.702 | 0.711 | 0.700 | 0.693 | 0.706 | 0.695 | 0.696 | 0.709 | 0.697 |
| DBA(g=2) | 0.767 | 0.768 | 0.766 | 0.750 | 0.752 | 0.751 | 0.754 | 0.757 | 0.755 | 0.741 | 0.747 | 0.746 |
| DBA(g=8) | **0.812** | **0.812** | **0.812** | **0.807** | **0.806** | **0.806** | **0.823** | **0.823** | **0.823** | **0.790** | **0.790** | **0.790** |

*Stage 1: Pretrain-member Separation via Cosine Similarity.* Given a sample $x = \{x_{\text{img}}, x_{\text{qst}}, x_{\text{ans}}\}$, we query the target VLM $f_\theta$ to obtain its response $y$. Using a sentence embedding model $f(\cdot)$, we compute the cosine similarity $\text{CS}(f(x_{\text{ans}}), f(y))$. If the similarity is below a threshold $\tau$, the sample is predicted as a *Pretrain-member*:

$$\text{If } \text{CS}(f(x_{\text{ans}}), f(y)) < \tau, \quad M(x) = \text{Pretrain-member}$$

*Stage 2: Finetune-member vs. Non-member via RIGEL.* If the sample is not classified as a Pre-member, we proceed to extract the RIGEL features—per-token generation time $T$, reported confidence $C$, and embedding length $L$—as defined previously. A binary classifier $\mathcal{C}_{\text{RIGEL}}$ is then used to classify the sample into Finetune-member or Non-member.

Compared to the unified multi-class strategy, DB-MIA enables tighter control over the classification boundary at each phase, especially under skewed data distributions Tramèr et al. (2022). Crucially, DB-MIA demonstrates high robustness in challenging scenarios where cosine similarity alone fails to reliably distinguish between finetuning members and non-members.

## 5 EXPERIMENT

### 5.1 EVALUATION SETTING

We evaluate our approach across four widely-used large-scale Vision-Language Models (VLMs) to assess attack robustness under diverse pretraining and finetuning paradigms. Our testbed includes: a) two LLaVA variants: `LLaVA-v1.5-7b-vicuna` and `LLaVA-v1.5-7b-llama-chat`; b) Two Qwen models: `Qwen2-vl-7b` and `Qwen2-vl-2b`. This selection provides architectural diversity through distinct backbone language models (Vicuna, LLaMA-2-Chat, and Qwen2) and implementation differences, enabling comprehensive evaluation of attack generalization. For each target model, we performed finetuning using its official pretraining checkpoint and a unified dataset. This dataset is derived from the LLaVA project's publicly released `llava_instruct_150k`, containing approximately 158,000 high-quality multimodal samples. Each sample consists of an image and a GPT-4-generated (instruction, answer) pair. The dataset covers broad capabilities including commonsense reasoning, factual knowledge, visual understanding, and dialogue-based instruction following. We partitioned the dataset into two disjoint subsets using a realistic 80:20 ratio: 1) $D_{\text{finetune}}$: used for LoRA-based parameter-efficient finetuning of VLMs; 2) $D_{\text{non}}$: held out entirely during finetuning, serving as the non-member reference for evaluation. All models were finetuned under identical hyperparameter settings (learning rate, batch size, epochs, LoRA rank, etc.). We monitored validation loss to ensure convergence and prevent overfitting, which rule-out the effects of finetuning membership and ensures rigorous attack evaluation.

### 5.2 OVERALL RESULTS

We evaluate the effectiveness of our proposed membership inference attacks (MIAs) through comprehensive experiments across the four state-of-the-art VLMs: LLaVA-v1.5-7b-vicuna, LLaVA-v1.5-7b-llama-chat, Qwen2-vl-7b and Qwen2-vl-2b. Each model undergoes three distinct attack strategies: Cosine Similarity Attack (CSA), Multi-class MIA, and Dual-Binary Attack (DBA). Table 1 compares precision, recall, and accuracy across all attack settings. These metrics are reported as macro-averages over pretraining members, finetuning members, and non-members. The Cosine Similarity Attack (CSA) demonstrates the weakest performance, with accuracy scores consistently ranging between 0.53–0.56 across all models. As a threshold-based method relying exclusively on cosine similarity between generated answers and ground truth, CSA exhibits limited discriminative capacity – particularly in distinguishing pretraining members from non-members. This limitation

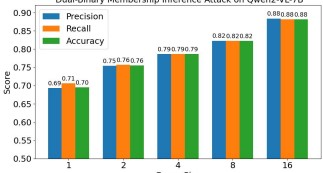 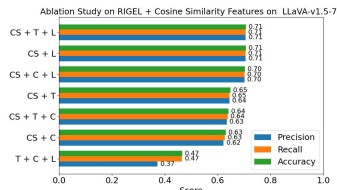

Figure 4: Effect of group size $g$ on MCA performance.

Figure 5: Effect of group size $g$ on DBA performance.

Figure 6: Ablation study on different feature combinations.

indicates that while VLMs do exhibit memorization patterns, these signatures remain insufficiently distinct when measured through a single similarity metric. The Multi-class Attack (MCA) demonstrates significant improvement over CSA, achieving accuracy scores of 0.67–0.68. This performance gain stems from integrating complementary behavioral features through our proposed RIGEL score—specifically generation time, response confidence, and embedding length. By capturing these multidimensional signals, MCA effectively identifies nuanced differences in VLM behavior when processing familiar versus unfamiliar samples. Building on this improvement, the Dual-Classifier Attack (DCA) reformulates the problem as a two-stage classification task. When grouping inputs into pairs ($g = 2$) and aggregating their features, DCA achieves approximately 0.75 accuracy across models. This grouping strategy reduces prediction variance and stabilizes outputs against stochastic generation effects—particularly valuable under randomness-inducing configurations like temperature sampling. The proposed Dual-Binary Attack (DBA) achieves the optimal performance, attaining accuracy scores exceeding 0.80 across all evaluated VLMs. DBA employs a hierarchical approach: first isolating pretraining-members using cosine similarity, then distinguishing finetuning-members from non-members through RIGEL-based features. This two-stage structure intentionally mirrors the VLM training pipeline, enabling simultaneous capture of coarse memorization patterns and fine-grained task-specific overfitting. The method achieves peak performance (0.823 accuracy) on Qwen2-VL-7B, representing the highest accuracy in our evaluation framework.

In addition, we evaluate the MaxRényi-K% metric, which measures the maximum Rényi uncertainty over the top-$K$ probability mass during sequence generation Hu et al. (2025). For this analysis, we focus on the combination of $\alpha = 5$ and $K = 1\%$, which was observed to yield relatively better discriminative performance in our preliminary experiments. The MaxRényi-K% metric captures subtle differences in the output probability distributions of VLMs, reflecting memorization patterns during text generation. As shown in Table 1, the MaxRényi-K% method exhibits moderate performance when distinguishing among the three member categories, with precision and recall around 0.39–0.42. In fact, the method performs reasonably well when distinguishing only finetuning-members from non-members. However, its effectiveness drops sharply once pretraining members—whose Rényi values are similar to those of finetuning-members—are included in the classification task. These observations indicate that while MaxRényi-K% can capture task-specific overfitting, it is less capable of distinguishing across VLMs training lifecycle.

These results validate our central hypothesis: The interplay of model overfitting, generation dynamics, and runtime behavior encodes sufficient information for effective multi-class membership inference for VLMs. The progressive performance gains from CSA to MCA to DBA demonstrate that DP-MIA can achieve robust inference due to enriched feature engineering (RIGEL) and tailored hierarchical prediction frameworks (DBA).

### 5.3 GROUP-BASED INFERENCE ANALYSIS

Same as in real attack scenarios, we evaluated the batched processing—testing groups of samples rather than individual inputs. This reflects practical conditions where attackers analyze query batches to enhance inference robustness through aggregated analysis. We systematically investigate group size $g$ as a key parameter for variance reduction. Grouping mitigates prediction instability under noisy generation conditions, particularly for stochastic VLM outputs. Figure 4 and Figure 5 show performance trends for MCA and DBA across group sizes on Qwen2-VL-7B (representative of consistent patterns across all VLMs). For MCA, accuracy steadily improves from 0.67 ($g = 1$) to 0.82 ($g = 16$), confirming that multi-sample aggregation suppresses generation noise and enhances behavioral consistency. Notably, gains saturate beyond $g = 4$, with marginal improvements between $g = 8$ and $g = 16$. DBA exhibits more substantial improvements, with accuracy rising

sharply from 0.70 $g = 1$ to 0.88 at $g = 16$. This demonstrates how DBA's hierarchical architecture, which includes coarse separation of pre-training members via cosine similarity followed by fine-grained RIGEL-based classification, amplifies the noise-averaging benefits of grouping. The dual-layer structure proves especially effective when distinguishing fine-tuning members from non-members. These results proves that sample grouping is a simple yet powerful mechanism for enhancing membership inference.

### 5.4 ABLATION STUDY ON FEATURE CONTRIBUTIONS

We conduct an ablation study to quantify the impact of individual features. The analysis evaluates combinations of four core components: cosine similarity (CS), average generation time per embedding (T), model confidence score (C), and generated response length (L). All experiments employ Logistic Regression for three-class classification (pretraining-members, finetuning-members, non-members) on the LLaVA-7B model, evaluated through macro-averaged precision, recall, and accuracy. Figure 6 presents results for seven key feature combinations. Among simpler configurations, CS+C and CS+L demonstrate moderately competitive performance, indicating that semantic confidence or response verbosity effectively complements similarity measures in detecting overfitting. The standalone RIGEL subset (T+C+L) achieves respectable metrics, confirming its intrinsic discriminative power without cosine similarity. Most significantly, combinations integrating CS with all three RIGEL components—CS+T+C, CS+T+L, and CS+C+L—consistently outperform other settings. The CS+T+L configuration achieves peak accuracy, revealing that both semantic alignment (CS) and generation dynamics (T,L) are essential. This synergy highlights the complementary nature of these features: where similarity captures memorization patterns, timing and verbosity expose behavioral signatures during inference. These findings confirm that DP-MIA's composite semantic alignment metrics is required for robust membership inference on VLMs.

## 6 DISCUSSION

**Feasibility of traditional defenses**: Traditional defense mechanisms—including Differential Privacy (DP) Dwork et al. (2006), data augmentation Shorten & Khoshgoftaar (2019), and regularization techniques like dropout and early stopping—remain viable for reducing overfitting and memorization. However, their application to VLMs are hindered by nontrivial trade-offs between computational overhead and model utility due to VLMs architecture's scale and complexity.

**Countering composite feature**: RIGEL leverages a wide spectrum of features across generation time, confidence scores and response length, which is harder to counter than single feature design. Our study shed light on potential counter measures: 1) output sanitization: limiting confidence score granularity; 2) response regulation: controlling answer length variability; 3) runtime obfuscation: introducing randomized processing delays. While these techniques could obscure membership cues, their practical efficacy and implementation costs require rigorous evaluation.

**Potential defense against DP-MIA**: Defending multi-stage VLMs against DP-MIA remains an open research challenge. Our analysis suggests that effective mitigation will likely require integrated solutions combining training-time interventions (DP, adversarial regularization) and deployment-time safeguards (output/runtime controls). Still, such approach faces challenges in establishing optimal privacy-utility tradeoffs for real-world VLM deployments.

## 7 CONCLUSION

This paper presents DP-MIA, the first multi-class MIAs against Vision-Language Models (VLMs) that distinguishing pretraining-member, finetuning-member, and non-member. To address binary MIA limitations, DP-MIA systhesizes the Cosine Similarity Attack (CSA) baseline, the composite metric RIGEL (runtime, confidence, and output length features), and the hierarchical Dual-Binary Attack (DBA) aligned with VLMs training paradigms. Experiments on SOTA VLMs validate DP-MIA's effectiveness, feature complementarity and robustness. Our findings enable novel training data accountability tracing for VLMs, while also reveal potential VLMs privacy vulnerabilities and shed light on multi-phase defenses in the future.

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

# A APPENDICES

## A.1 THE USE OF LARGE LANGUAGE MODELS

We used Large Language Models in paper writing only to aid or polish writing of some paragraphs. We did not use Large Language Models in research ideation or writing to the extent that they could be regarded as a contributor.

## A.2 REPRODUCIBILITY STATEMENT

This paper strictly follows the reproducibility standards of ICLR. We publicly released the code and experiment results at `https://github.com/frozen-jak/Dual-Phase-Mia` without revealing the authors' identity.

