# OpenReview forum: "DP-MIA: DUAL-PHASE MEMBERSHIP INFERENCE ATTACK ACROSS VLMS TRAINING LIFECYCLE"
_ICLR.cc/2026/Conference — ICLR 2026 Conference Withdrawn Submission_

### Official Review · Reviewer_9EBg · 2025-10-27

**Soundness:** 3
**Presentation:** 3
**Contribution:** 3
**Rating:** 6
**Confidence:** 3

**Summary:**

The paper formulates membership inference for vision–language models (VLMs) as a three-way attribution problem—pretraining member, finetuning member, or non-member—and proposes a black-box framework, DP-MIA, to solve it. It introduces a composite signal, RIGEL (response time, self-reported confidence, and generated output length), plus a hierarchical Dual-Binary Attack (DBA) that first separates pretraining members by cosine similarity and then distinguishes finetuning members from non-members using RIGEL. Experiments on LLaVA and Qwen2-VL variants report up to 88.2% accuracy, with grouping multiple queries further improving robustness. The work targets production-like black-box settings and releases code to facilitate reproducibility.

**Strengths:**

1.Clear problem motivation and scope: auditing which training stage (pretrain vs. finetune) exposed a sample is practically important yet under-explored in VLMs; most recent VLM-MIA studies focus on binary MIAs (often for instruction-tuning) or single-modality CLIP-style settings. Relative to these, a tri-class, stage-aware formulation fills a gap.

2.Black-box emphasis is timely: unlike methods that assume logit access (e.g., MaxRényi-K%), DP-MIA’s signals are observable from typical API outputs (text, latency, self-reported confidence), broadening applicability.

3.The hierarchical DBA design is well aligned with the two-stage VLM pipeline (contrastively pre-trained, then instruction-tuned), and the paper shows that cosine similarity suffices for coarse separation before using richer features

4.Group-based inference is a reasonable variance-reduction strategy that the broader literature has found effective (e.g., aggregation across transformations or small sets), and the paper shows tangible gains with larger group sizes.

**Weaknesses:**

1.MaxRényi-K% (NeurIPS 2024) typically uses logit access (a grey-box assumption), whereas DP-MIA stresses black-box outputs; separating black-box from logit-access baselines (or providing a logit-free Rényi proxy) would be a fairer comparison.

2.Ground-truth “pretraining membership” is partially credible only for models with released pretraining sets (e.g., LLaVA’s 558K LAION/CC/SBU subset), but less so for models whose pretraining corpora are undisclosed or vast (e.g., Qwen2/2.5-VL report scaled trillions of tokens without full release), risking label noise; simulating both stages end-to-end on known data would tighten validation.

3.The “generated embedding length” L is insufficiently specified (e.g., whether it is token count or a sum over per-token embedding norms from a specific encoder); a clearer definition and ablations vs. simple token-length baselines are needed.

4.Editorial issues slightly hinder readability, for example, the “Consine Similarity” in Fig.2,  the “Daul-Binary” in Fig.3.

**Questions:**

Please refer to the Weakness section for corresponding questions.

---

### Official Review · Reviewer_HAsT · 2025-10-29

**Soundness:** 2
**Presentation:** 3
**Contribution:** 2
**Rating:** 4
**Confidence:** 5

**Summary:**

The paper introduces DP-MIA, a new multi-class membership inference attack (MIA) framework targeting multi-stage Vision-Language Models (VLMs). Unlike conventional MIAs that perform binary classification (member vs. non-member), DP-MIA distinguishes among three exposure states: Pretraining member, Finetuning member, Non-member
This distinction is motivated by the multi-stage training lifecycle of modern VLMs, where different data sources (and ownerships) contribute to pretraining and finetuning. The paper aims to enable fine-grained accountability tracing and privacy auditing in black-box settings—an underexplored but important problem.

**Strengths:**

This paper has excellent writing and makes it easy for the reader to follow.
And the author raises a really important question, which is how to distinguish whether exposure occurred during pretraining, finetuning, or not at all. This question is relevant to identifying which parties need to be punished for unauthorized use of training data.
Existing membership inference attacks typically consider only binary scenarios — distinguishing between pretraining members and non-members, or finetuning members and non-members. However, they fail to account for the realistic setting where all three categories coexist simultaneously. The author proposes a method to solve this question.

**Weaknesses:**

Although the problem addressed in this paper is very interesting, I find the proposed method somewhat lacking in novelty. The main goal of the paper is to distinguish among pretraining members, finetuning members, and non-members. However, the proposed DB-MIA essentially decomposes this task into two existing components: it first applies a cosine-similarity-based MIA to identify pretraining members, and then uses an existing Generated Embedding Length–based MIA to separate finetuning members from non-members (while the ablation study suggests that the other two proposed indicators contribute little to the final performance).

That said, one particularly intriguing observation is that finetuning members and non-members exhibit surprisingly similar cosine similarity scores, both of which are even higher than those of pretraining members. This result is counterintuitive—intuitively, both pretraining and finetuning members, being data the model has already seen, should yield higher cosine similarity than non-members. If the authors could provide a convincing explanation or theoretical analysis of why this counterintuitive behavior occurs, it would substantially strengthen the paper’s contribution. I would consider raising my overall score if this aspect were clarified and justified.

**Questions:**

1. Why finetuning members and non-members exhibit surprisingly similar cosine similarity scores, both of which are even higher than those of pretraining members?

2. From the ablation study, it seems like per-token generation time and confidence are not necessary for the method. Please justify that.

---

### Official Review · Reviewer_RPUU · 2025-10-29

**Soundness:** 2
**Presentation:** 3
**Contribution:** 2
**Rating:** 2
**Confidence:** 3

**Summary:**

This paper presents a membership inference attack framework for multi-stage VLMs, aiming to distinguish whether data exposure occurred during pretraining, finetuning, or not at all. The approach introduces techniques such as the RIGEL composite metric, group-based inference, and a Dual-Binary Attack design, addressing challenges of black-box access and weak memorization signals. Experimental results on LLaVA and Qwen2-VL demonstrate strong performance and clear advantages over existing MIAs.

**Strengths:**

1.	The paper proposes a well-motivated framework that moves beyond traditional binary MIAs to achieve fine-grained phase attribution in VLMs. They proposed the RIGEL composite metric and hierarchical attack design demonstrates strong creativity and clear practical relevance for privacy auditing in black-box models.
2.	This paper is well-structured and with potential defence discussion.

**Weaknesses:**

1.	The proposed composite metric RIGEL includes model response time as a feature, but inference time is influenced by many external factors (e.g., network latency, system load, and model generation strategies). These factors make it unstable and not directly related to the memorization phenomenon. Moreover, longer response times may simply reflect the model’s effort to generate higher-quality or more coherent outputs rather than indicating memorization. Therefore, I think that using response time as part of the metric may not be a reliable or theoretically sound choice.
2.	The proposed use of self-reported confidence scores also raises concerns about the reliability of the RIGEL composite metric.

3.	While the paper emphasizes distinguishing between pretraining and finetuning exposure as a key advancement, this distinction may be somewhat overstated. The fundamental goal of membership inference is to evaluate a model’s vulnerability to privacy leakage, regardless of which stage the memorization occurred. In practice, if a model is prone to leaking training data after pretraining, it is likely to remain vulnerable after finetuning or other task-specific adaptations. Therefore, the proposed phase-specific differentiation, though interesting, may not substantially enhance the broader understanding of model privacy risks, and could be seen as a rather narrow or niche extension of prior work.

4.	The paper refers to “recent work” on black-box attacks by Yeom et al. (2018) and Salem et al. (2018), but these studies are relatively dated in the context of rapid developments in privacy and adversarial research.

5.	The proposed Cosine Similarity Attack (CSA) relies on measuring similarity between the model’s generated response and a “ground-truth” answer using cosine similarity. However, VLM outputs are inherently generative and open-ended, unlike deterministic classification tasks with clear labels. Therefore, cosine similarity may not reliably indicate correctness or membership, as it can be heavily affected by surface-level wording and by the semantic representation capability of the chosen feature encoder. A more comprehensive discussion and comparison of different embedding models and alternative similarity metrics would strengthen this part of the methodology, as well as potentially reveal more robust measures of model memorization.

6.	The authors assign a higher weight to output length in the RIGEL score but do not explain why it should be more important than runtime or confidence. This makes the weighting choice seem arbitrary and less justified.

**Questions:**

please check the weakness

---

### Official Review · Reviewer_fGHz · 2025-11-01

**Soundness:** 1
**Presentation:** 2
**Contribution:** 2
**Rating:** 2
**Confidence:** 3

**Summary:**

The paper proposes a new membership inference attack (MIA) framework for VLMs. Unlike existing MIA methods that do not differentiate whether a data point belongs to the pretraining or fine-tuning stage, the authors introduce a more fine-grained categorization: pre-member, fine-member, and non-member. To address this problem, they propose three strategies: (1) Cosine Similarity Attack, which determines the membership of a data point based on the cosine similarity between the model output and the ground truth; (2) RIGEL, a black-box metric that predicts membership using response time, output confidence, and output length; and (3) Dual-Binary Membership Inference Attack (DB-MIA), which combines the strengths of (1) and (2). Experiments conducted on four VLMs demonstrate the effectiveness of these strategies in identifying membership across fine-grained categories.

**Strengths:**

- Fine-grained categorization for membership inference attacks in VLMs is an important problem, as these models are typically trained in multiple stages.

- Experimental results demonstrate the effectiveness of the three proposed strategies in identifying membership across these fine-grained categories.

**Weaknesses:**

- Motivation for the three proposed strategies is unclear. For example, the RIGEL score is computed using Per-embedding Generation Time, Confidence Score, and Generated Embedding Length. While these features are intuitively reasonable, the paper lacks empirical evidence or theoretical justification explaining why they are important for identifying data membership.

- The proposed strategies heavily rely on multiple thresholds. Moreover, the paper lacks experiments analyzing how these thresholds affect the results.

- Section 5.2 reports group-based attack results only using the third strategy (DB-MIA). It remains unclear whether group-based attacks could also improve the accuracy of the first two strategies. Further ablation studies on group size are needed to investigate the effect of using a batch of inputs.

- The paper lacks of comparison with existing MIAs. Although traditional MIAs only distinguish between member and non-member, a simple comparison could be made by treating both pre-member and fine-member as members. Providing such a comparison would help demonstrate the effectiveness of the proposed methods relative to existing approaches.

- The related work omits recent membership inference attacks in DNNs, LLMs, and VLMs (e.g., [1–3]). Authors may consider to update the related work with recent MIAs.


[1] Membership Inference Attacks against Large Vision-Language Models, Neurips 2024

[2] Low-cost high-power membership inference attacks, ICML 2024

[3] Not All Tokens Are Equal: Membership Inference Attacks Against Fine-tuned Language Models, ACSAC 2024.

**Questions:**

- Could the authors provide a justification for why the proposed strategies are effective for fine-grained membership inference attacks?

- Could the authors include a comparison with recent membership inference attacks in VLMs?

- Could the authors evaluate the effect of group size on the other two proposed strategies and analyze how group size impacts attack accuracy?

---

### Note · Authors · 2025-11-14

I have read and agree with the venue's withdrawal policy on behalf of myself and my co-authors.